# Evaluation of Prognostic Parameters to Identify Aggressive Penile Carcinomas

**DOI:** 10.3390/cancers15194748

**Published:** 2023-09-27

**Authors:** Jan Niklas Mink, Oybek Khalmurzaev, Alexey Pryalukhin, Carol Immanuel Geppert, Stefan Lohse, Kristof Bende, João Lobo, Rui Henrique, Hagen Loertzer, Joachim Steffens, Carmen Jerónimo, Heiko Wunderlich, Julia Heinzelbecker, Rainer M. Bohle, Michael Stöckle, Vsevolod Matveev, Arndt Hartmann, Kerstin Junker

**Affiliations:** 1Department of Urology and Paediatric Urology, Saarland University, 66421 Homburg, Germanymichael.stoeckle@uks.eu (M.S.);; 2Department of Urology, Federal State Budgetary Institution “N.N. Blokhin National Medical Research Center of Oncology”, Ministry of Health of the Russian Federation, Moscow 115478, Russia; 3Institute of Pathology, Saarland University Medical Centre, 66421 Homburg, Germany; 4Institute of Pathology, University Erlangen-Nuremberg, 91054 Erlangen, Germany; 5Institute of Virology, Saarland University, 66123 Homburg, Germany; 6Department of Pathology and Cancer Biology and Epigenetics Group—Research Center, Portuguese Oncology Institute of Porto/Porto Comprehensive Cancer Center Raquel Seruca, School of Medicine and Biomedical Sciences (ICBAS), University of Porto, 4050-513 Porto, Portugal; jpedro.lobo@ipoporto.min-saude.pt (J.L.); henrique@ipoporto.min-saude.pt (R.H.);; 7Clinic of Urology and Paediatric Urology, Westpfalz-Klinikum, 67655 Kaiserslautern, Germany; 8Department of Urology and Paediatric Urology, St. Antonius Hospital, 52249 Eschweiler, Germany; 9Clinic of Urology and Paediatric Urology, St. Georg Klinikum, 99817 Eisenach, Germany

**Keywords:** penile cancer, HPV, p16, prognosis, histological subtype

## Abstract

**Simple Summary:**

Sufficient prognostic parameters are still lacking in penile cancer. In this study, we sought to evaluate the current TNM classification in terms of its ability to estimate prognosis and to identify additional independent prognostic parameters. We found that lymph node metastasis—as well as lymphovascular invasion in node-negative patients—had the strongest impact on prognosis, whereas HPV did not show an influence on outcome. Furthermore, the pT1b stage seems questionable, and a revision of the current TNM classification is advised.

**Abstract:**

Background: Advanced penile carcinoma is characterized by poor prognosis. Most data on prognostic factors are based on small study cohorts, and even meta-analyses are limited in patient numbers. Therefore, there is still a lack of evidence for clinical decisions. In addition, the most recent TNM classification is questionable; in line with previous studies, we found that it has not improved prognosis estimation. Methods: We evaluated 297 patients from Germany, Russia, and Portugal. Tissue samples from 233 patients were re-analyzed by two experienced pathologists. HPV status, p16, and histopathological parameters were evaluated for all patients. Results: Advanced lymph node metastases (N2, N3) were highly significantly associated with reductions in metastasis-free (MFS), cancer-specific (CS), and overall survival (OS) rates (*p* = <0.001), while lymphovascular invasion was a significant parameter for reduced CS and OS (*p* = 0.005; *p* = 0.007). Concerning the primary tumor stage, a significant difference in MFS was found only between pT1b and pT1a (*p* = 0.017), whereas CS and OS did not significantly differ between T categories. In patients without lymph node metastasis at the time of primary diagnosis, lymphovascular invasion was a significant prognostic parameter for lower MFS (*p* = 0.032). Histological subtypes differed in prognosis, with the worst outcome in basaloid carcinomas, but without statistical significance. HPV status was not associated with prognosis, either in the total cohort or in the usual type alone. Conclusion: Lymphatic involvement has the highest impact on prognosis in penile cancer, whereas HPV status alone is not suitable as a prognostic parameter. The pT1b stage, which includes grading, as well as lymphovascular and perineural invasion in the T stage, seems questionable; a revision of the TNM classification is therefore required.

## 1. Introduction

Penile carcinoma (PC) is a rare tumor disease with an incidence of 1/100,000 men in western Europe, while the estimated incidence in developing countries is much higher with up to 10% of all malignant diseases in men [1,2,3]. Due to the low incidence of the disease, there is a lack of large patient cohorts, and even meta-analyses have been limited in patient numbers. PC mostly occurs in elderly men, with an age peak between 60 and 70 years [4]. In addition to poor hygiene conditions and phimosis, infection with high-risk HPV subtype (hrHPV) is the most important risk factor for the development of PC, occurring in 30–50% of cases [5,6]. Of those hrHPV types, HPV16 has been identified as the predominant type based on a meta-analysis [6].

Active expression of functional hrHPV viral oncoproteins is causally linked to the malignant transformation of proliferating cells [7]. Detection of viral DNA in cancer specimens can indicate productive, transformed, latent, or silent infection [8]. To reliably identify HPV-positive cancers with active viral oncoproteins, HPV status is determined by a combination of viral DNA-PCR testing and p16^INK4a^ immunohistochemistry because p16^INK4^a is the established surrogate marker for HPV-driven transformation [9,10,11]. This HPV status definition can substantially improve the accuracy of estimating clinical outcomes for cancer patients with HPV-associated cancers [7,12].

The majority of PCs are penile squamous cell carcinomas (PSCCs) with different histological subtypes. Histological subtyping of penile carcinomas is based on their HPV status, so that there are HPV-associated and non-HPV-associated subtypes. The most common histological subtype is the usual PSCC, which accounts for about 75% of all cases and is mostly HPV negative, followed by warty-basaloid, basaloid, and warty PSCCs, the latter being strongly associated with HPV-induced carcinogenesis [13]. The histological subtypes significantly differ in their aggressiveness and probability of metastasis [14,15], which is why an evaluation of their prognostic value would appear to be useful; however, sufficient data are still lacking.

The TNM classification is used to estimate prognosis. Because the previous version was found to be inadequate, an adjustment of the TNM classification was made in its eighth edition [16]; however, this adjustment has not led to relevant improvement in estimating prognosis [17], so that there is still a lack of sufficient parameters for prognosis estimation.

For this reason, in this study, we investigated the current TNM classification as well as other putative risk factors with the aim of providing a more accurate prognosis estimation and thus reducing the risk of over- or undertreatment.

## 2. Materials and Methods

### 2.1. Cohort and Study Design

A total of 297 patients from multiple centers in Germany (46 from the University of Saarland, Homburg; 34 from Helios Clinic, Erfurt; 21 from St. Antonius Hospital, Eschweiler; 12 from St. Georg Hospital, Eisenach; 10 from Westpfalz Hospital, Kaiserslautern; and 6 from Helios Clinic, Bad Blankenheim), Russia (128 from N.N. Blokhin National Research Center of Onkology, Moskow), and Portugal (40 from the Portuguese Oncology Institute of Porto), who had been treated for penile carcinoma between 1989 and 2018, were included in this cohort. The Saarland ethical committee confirmed analyses of patient data and tumor samples. Representative formalin-fixed and paraffin-embedded tissue samples from 233 patients were analyzed. Experienced uropathologists from two German university centers reviewed all tissue samples and re-examined the respective histological subtypes, as well as lymphovascular, vascular, and perineural invasions, according to the 2016 WHO classification [13]. All tumors were reclassified according to the most recent (eighth) edition of the TNM classification of malignant tumors [16].

Tissue microarray construction and immunohistochemistry with an evaluation of p16*^INK4a^* were performed, as previously reported, using a monoclonal antibody against p16*^INK4a^* (Abcam, clone 1D7D2A1, Boston, MA, USA), with 1:4000 dilution [17]. Tumor samples from the center, tumor front, and lymph node metastases with a spot size of 1.5 mm each were used for tissue microarray construction.

DNA was isolated from FFPE tissue sections using a QIAamp DNA FFPE Tissue Kit (Qiagen, Cat. No. 56404, Hilden, Germany) according to the manufacturer’s protocol. HPV PCR [18] was performed with GP5+/6+ primers (final concentration 0.5 µM) using LightCycler 1.5 (Roche Diagnostics GmbH, Basel, Switzerland) and LightCycler FastStart DNA Master Plus SYBR Green I (Roche Diagnostics GmbH, Cat. No. 03515885001). Initial denaturation at 95 °C for 15 min was followed by 45 cycles of PCR with denaturation at 95 °C for 10 s, primer hybridization at 45 °C for 5 s, and elongation at 72 °C for 18 s. HPV16 and HPV18 DNA were included as positive controls. In parallel, GAPDH PCR (housekeeping gene) was performed for the detection of cellular DNA. Amplicons were then separated on an agarose gel (3%) and documented using ethidium bromide and a BIO-RAD ChemiDoc XRS+ system. For genotyping, the amplified DNA was sequenced via seq-it GmbH & Co.KG (Kaiserslautern, Germany) and analyzed using the Basic Local Alignment Search Tool (Version 2.11.0, BLAST, NCBI).

### 2.2. Statistical Analyses

Statistical analyses were performed using SPSS Statistics 28 (Statistical Package for Social Science, IBM^®^, Armonk, NY, USA).

The Kaplan–Meier estimator was used to calculate survival curves, and the log-rank test was used to test the statistical significance. Uni- and multivariable analyses were performed using the Cox proportional hazard model. A *p* value < 0.05 was considered as statistically significant. All significant parameters in univariable analysis were further investigated in a multiple analysis.

## 3. Results

### 3.1. Patient Characteristics:

The median patient age was 63 years (range 24–93), with mean ages of 58, 67, and 77.5 years among Russian, German, and Portuguese patients, respectively. HPV status could be evaluated in 222 patients. Among these, 79 tumor samples (35.6%) were positive for hrHPV-DNA. In addition, HPV subtype 16 was found in 72 tumor samples, subtypes 18 and 35 in 2 samples, and subtype 59 in 1 sample. In two tumor samples, the exact subtype could not be identified. p16*^INK4a^* was detected in 69 of these tumors (87.3%). Thus, 31.1% of the specimens displayed an HPV-positive status based on DNA detection and p16 staining (33.6% from Russia, 29.9% from Germany, and 25.6% from Portugal). 

Comparing histological subtypes (Table 1), we found that the usual type was the most common subtype at 53.3%, with 21.1% of usual-type tumors being HPV positive. The most common HPV-associated tumors were warty-basaloid (13.3%, 65.7% HPV positive), basaloid (11.4%, 77.8% HPV positive), and warty carcinomas (5.7%, 20% HPV positive).

After comparing the seventh and eighth editions of the TNM classification systems, one patient was shifted from the pT1a to the pT1b stage due to the inclusion of perineural invasion in the pT1b stage (Table 2) in the eighth edition. In addition, based on the differentiation between infiltration of the corpus spongiosum and cavernosum described in the eighth edition, 10 patients were upstaged from pT2 (seventh edition: 35.9%; eighth edition: 31.9%) to pT3 tumors (seventh edition: 20.6%; eighth edition: 25.0%). Overall, 31.3% of patients had lymph node metastasis at the time of diagnosis, with no changes between the seventh and eighth editions, whereas 17.9% had lymphovascular invasion, 21.4% vascular invasion, and 18.8% perineural invasion.

### 3.2. Survival Analysis

The median follow-up time was 27 months (range 3–253 months). Warty tumors had the best MFS, CSS, and OS (Table 3, Figure 1) rates. In contrast, basaloid tumors had the worst outcomes in CSS and OS; however, these findings were not statistically significant (log rank, *p* = 0.14 for CSS; *p* = 0.133 for OS).

Analysis of HPV-positive and HPV-negative tumors did not reveal any significant differences in MFS, CSS, and OS between the two groups, either in the total cohort (Table 3, Figure 2a–c) or in the subgroup analysis of usual-type PC (Figure 3a–c).

The T stage was significantly associated with MFS (log rank, *p* = 0.003), CSS (log rank, *p* = 0.006), and OS (log rank, *p* = 0.010). The pT1b stage had the worst five-year survival rate (44%) even when compared with pT2 (78%), pT3 (56%), and pT4 (50%) tumors. Five-year CSS and OS survival rates significantly decreased with increasing T stages (Table 3, Figure 4a–c).

Lymph node status was a highly significant parameter for survival. Increasing numbers of lymph node metastases resulted in reduced MFS, CSS, and OS (log rank, *p* = <0.001) (Table 3, Figure 5a–c).

Higher grades (G2–3) were associated with a significantly decreased MFS, CSS, and OS compared with G1 (MFS: log rank, *p* = 0.017; CSS: log rank, *p* = 0.034; OS: log rank, *p* = 0.025), but outcomes did not differ between G2 and G3 (Table 3, Figure 6a–c).

Vascular invasion was significantly associated with poorer OS rates. MFS and CSS were reduced in tumors with vascular invasion; however, this result was not statistically significant (Table 3, Figure 7a–c). 

CSS and OS were significantly reduced in patients with lymphovascular or perineural invasion. While patients with lymphovascular invasion also showed a significantly reduced MFS, the difference between perineural-negative and -positive patients was not significant (Table 3, Figure 8a–c and Figure 9a–c).

In the next step, Cox regression analysis was performed. In univariate analysis, multiple parameters correlated significantly with MFS, CSS, and OS (Table 4) values that were included in multivariate analysis (Table 5). Here, the pT1b stage (hazard ratio 7.8, *p* = 0.017) and advanced lymph node metastasis stages (pN2: hazard ratio 3.9, *p* = 0.007; pN3: hazard ratio 6.8, *p* = <0.001) were independent parameters to predict MFS. pN2 (hazard ratios: CSS 4.9, *p* = <0.001; OS 4.2, *p* = <0.001) and pN3 stages (hazard ratios: CSS 5.9, *p* = <0.001; OS 4.2, *p* = <0.001), as well as lymphovascular invasion (hazard ratios: CSS 2.7, *p* = 0.005; OS 2.3, *p* = 0.007) and the age at diagnosis, were independent prognostic parameters concerning CSS and OS. 

Finally, patients without lymph node involvement were separately investigated. In this subgroup, lymphovascular invasion was the only significant parameter to predict MFS (hazard ratio 2.8 (1.1–6.9), *p* = 0.032), whereas other prognostic factors included in the pT1b stage such as grading, vascular invasion, or perineural invasion did not show any significant differences in survival (Table 6, Figure 10a–c).

## 4. Discussion

Penile cancer remains a poorly studied tumor entity; as a result, prognosis in advanced stages also remains poor. Due to the low incidence of the disease, reliable systematic data on prognostic markers and systemic therapy are limited. Most studies have been based on small numbers of patients from single centers. In this study, we succeeded in recruiting an international multicenter patient cohort with a high number of patients to evaluate prognostic parameters; these are important when determining individualized therapeutic approaches, especially those with curative intention in the early stages of the disease. In brief, we found that HPV has no influence on prognosis, whereas lymph node metastasis and lymphovascular invasion are both significant independent prognostic parameters.

The frequency of HPV-related tumorigenesis was lower in our cohort than in several published data [6]. This might be due to the higher socioeconomic status of the countries from which patients were recruited for this study [19]. Interestingly, German and Portuguese patients have a comparatively low rate of HPV infection, while Russian patients tend to be younger, with a higher rate of HPV-associated tumors. These results underline the regional differences evident in our data. Nonetheless, HPV infection remains the most important factor in the development of penile carcinoma, along with poor hygiene conditions. This explains why countries with lower socioeconomic status continue to report a significantly higher incidence of penile carcinoma. Whether HPV vaccination will lead to a reduction in such incidence in the future remains to be seen. The long period between infection and the development of an HPV-driven PC means that there is a long lag time when observing promising trends of routine HPV vaccination programs [20]. 

The prognostic role of HPV status has not yet been clarified. In head and neck cancers, the data clearly demonstrate a better survival rate for HPV-induced tumors [21,22]; however, in penile cancer, the data remain unclear due to contradictory results [23,24,25,26,27,28]. Importantly, the better prognosis in HPV-positive head and neck tumors is not attributed to better differentiation or lower aggressiveness, but to increased sensitivity to chemotherapy and radiotherapy, both of which play a rather minor role in the curative treatment of penile cancer [21]. Furthermore, many studies that have shown a better prognosis for HPV-positive tumors in penile cancer have not involved p16^INK4a^ analysis; in such studies, therefore, the involvement of viral oncoproteins in tumorigenesis was not proved [26]. On the other hand, in the present study, HPV-related histological subtypes greatly differed with respect to prognosis: basaloid PC is characterized by very aggressive tumors, whereas the warty subtype is characterized by good prognosis and very low risk of metastasis [29]. This complicates any prognostic evaluation according to HPV status carried out independently of histological subtypes. Therefore, we separately analyzed usual-type carcinomas, as about 20–25% of these are HPV-related.

Our analysis did not reveal any association of HPV with prognosis, either in the total cohort or in the usual type alone. The latter analysis was performed to eliminate the bias of the deviating prognoses between histological subtypes. We may say, therefore, that HPV is not suitable as an independent prognostic parameter. However, histological subtypes differ in prognosis regardless of HPV status and should therefore be reported by pathologists and considered when making decisions concerning therapy. In our study, the number of other histological subtypes was too small to perform a meaningful analysis of the role of HPV. However, such an analysis, involving larger study cohorts, may be seen as a necessary future task.

Nevertheless, HPV status should still be determined, as future HPV-positive PC patients may benefit not only from targeted therapies but also from immunotherapies, as shown in the recently published study by de Vries et al. [30].

In line with the results of various previous studies, we found that the advanced lymph node metastasis stage (N2, N3) was the most important prognostic factor [31,32]. These results emphasize the importance of early lymph node management with complete inguinal and—if necessary—iliac lymph node resection. There is also an urgent need to develop new diagnostic tools for the evaluation of lymph node status, as metastases are underestimated in clinical examinations due to non-palpable micrometastases, which are already present in about 25% of patients at the time of diagnosis according to Borchers et al. [33]. 

Similarly, lymphovascular invasion was a significant parameter for shorter MFS in patients without proven lymph node metastases at the time of surgery. In line with previous studies, we found that the risk of distant metastasis significantly increases in the presence of lymphovascular invasion because infiltration of tumor cells into the lymphatic system is known to be a prerequisite for tumor spread [34,35,36]. Thus, lymphovascular invasion should always be considered in treatment planning as a potential marker for the presence of micrometastases, especially in node-negative patients, either clinically or image-morphologically. Such patients might also benefit from early lymphadenectomy or sentinel lymph node biopsy. Indeed, several studies have shown a significant survival benefit for patients with nonmetastatic high-risk PC when early lymphadenectomy is performed [37].

Based on our results, the inclusion of lymphovascular invasion, grading, and, starting from the eighth edition of TNM, perineural invasion into the classification of the primary tumor for the discrimination between pT1a and pT1b can be seen as unique within tumor staging guidelines and should be critically discussed. pT1b showed the worst outcome concerning MFS, such as non-organ-confined tumors. A better differentiation of risk factors in the TNM classification is necessary, considering lymphovascular invasion as an independent prognostic parameter to avoid overtreatment in low-malignant tumors, while aggressive tumors, especially with LVI, benefit from early aggressive therapy and a close follow-up. As the current TNM classification only partially improved its prognostic value, compared with the previous version [17,38,39], further revision should be considered. On the other hand, the most recent TNM classification did involve meaningful changes, such as the new categorization of pT2 and pT3 stages. In their meta-analysis of 3692 patients, Li et al. demonstrated a significantly better tumor-specific survival in the presence of corpus spongiosum infiltration compared with the infiltration of corpus cavernosum; therefore, this discrimination is of clinical relevance [40].

In addition to the retrospective character of this study, the low frequency of some histological subtypes may be seen as another limitation that prevented a more detailed analysis concerning prognostic evaluation. Further studies with higher patient numbers that focus on histological subtypes are therefore needed. Moreover, patients were evaluated over a period of almost 30 years; therefore, a wide variety of therapy concepts were applied that could not be considered in this study. Additional research is required to evaluate whether patients with lymph node metastases or lymphovascular infiltration may benefit from a more aggressive therapy.

## 5. Conclusions

HPV status plays an important role in the etiology of penile carcinoma; however, it is not associated with prognosis or metastatic potential. Additional markers beyond HPV status that more accurately reflect the underlying tumor biology may help improve prognostic estimates. In contrast, histological subtypes exert a major influence on prognosis; therefore, these should always be reported, with consequent impact upon therapy decisions.

Advanced lymph node metastasis and lymphovascular invasion in node-negative patients are the most important independent prognostic parameters. Therefore, an aggressive lymph node management is suggested for LVI-positive patients. Furthermore, the combination of lymphovascular invasion, grading, and perineural invasion into a single T category is arguably beneficial, and a more detailed discrimination of the risk factors in the TNM system appears to be urgently needed.

## Figures and Tables

**Figure 1 cancers-15-04748-f001:**
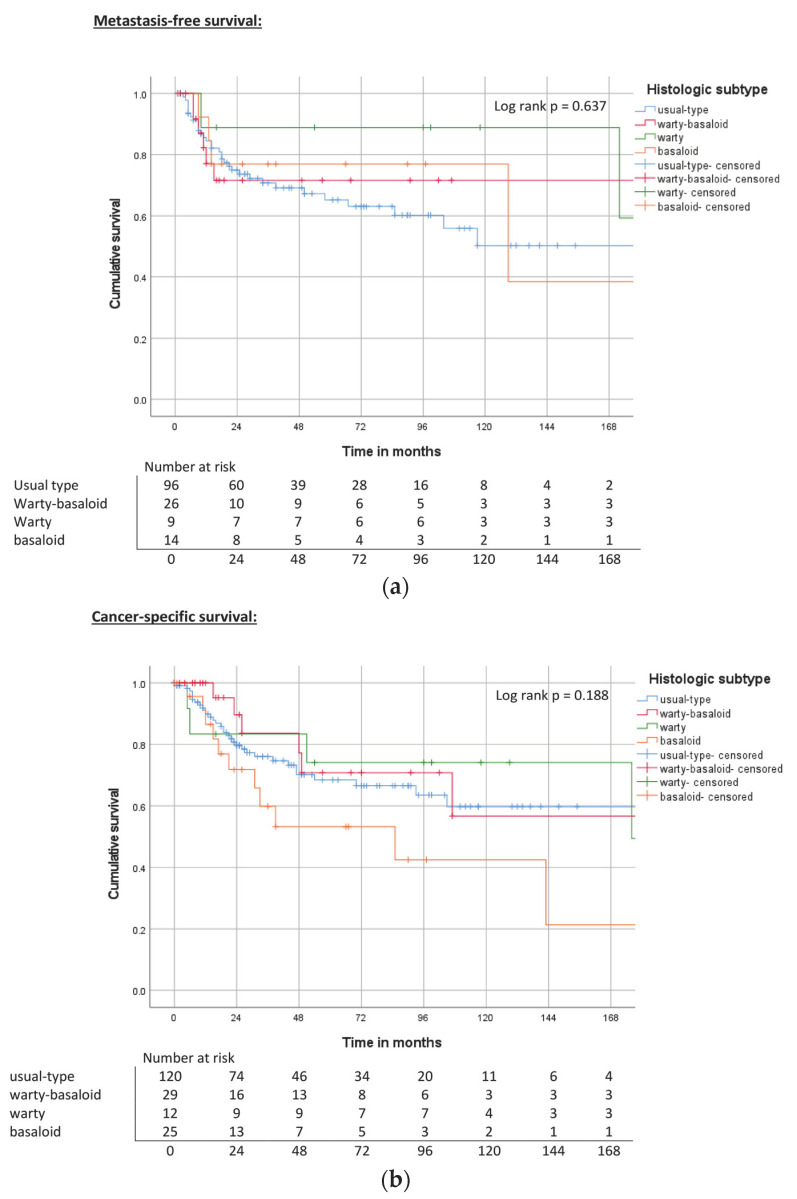
(**a**) Kaplan–Meier curve of MFS related to histological subtypes. (**b**) Kaplan–Meier curve of CSS related to histological subtypes. (**c**) Kaplan–Meier curve of OS related to histological subtypes.

**Figure 2 cancers-15-04748-f002:**
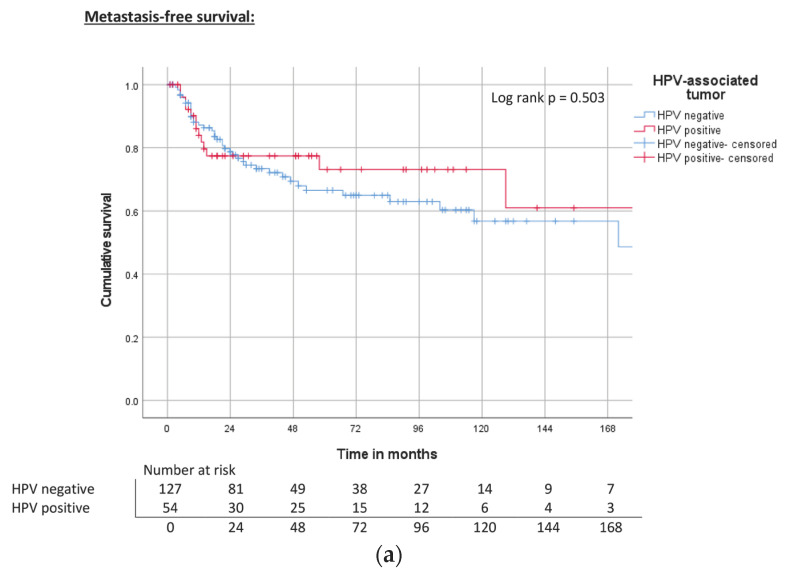
(**a**) Kaplan–Meier curve of MFS related to HPV status. (**b**) Kaplan–Meier curve of CSS related to HPV status. (**c**) Kaplan–Meier curve of OS related to HPV status.

**Figure 3 cancers-15-04748-f003:**
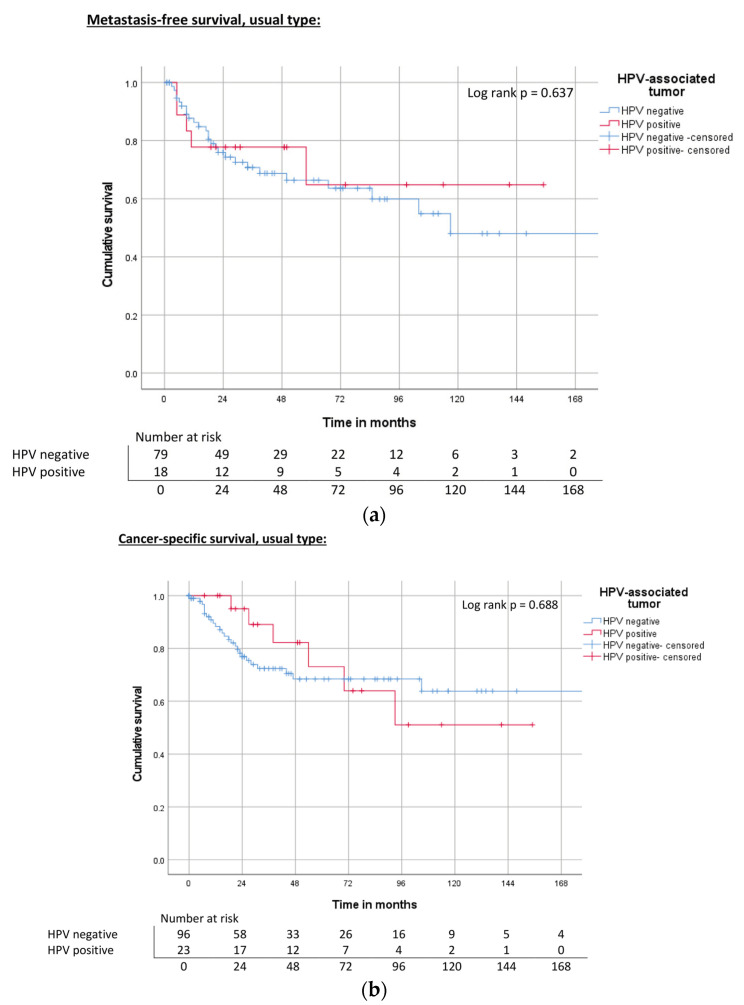
(**a**) Kaplan–Meier curve of MFS related to HPV status in the usual type. (**b**) Kaplan–Meier curve of CSS related to HPV status in the usual type. (**c**) Kaplan–Meier curve of OS related to HPV status in the usual type.

**Figure 4 cancers-15-04748-f004:**
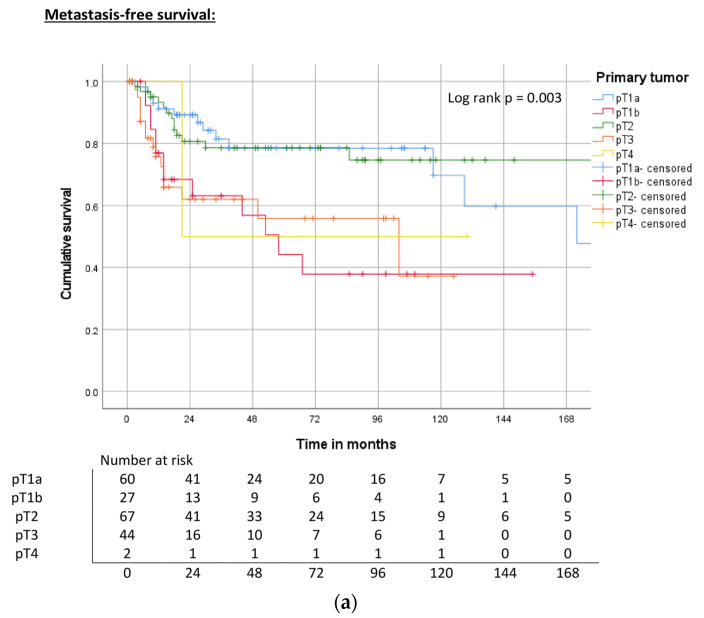
(**a**) Kaplan–Meier curve of MFS related to the T stage. (**b**) Kaplan–Meier curve of CSS related to T stage. (**c**) Kaplan–Meier curve of OS related to T stage.

**Figure 5 cancers-15-04748-f005:**
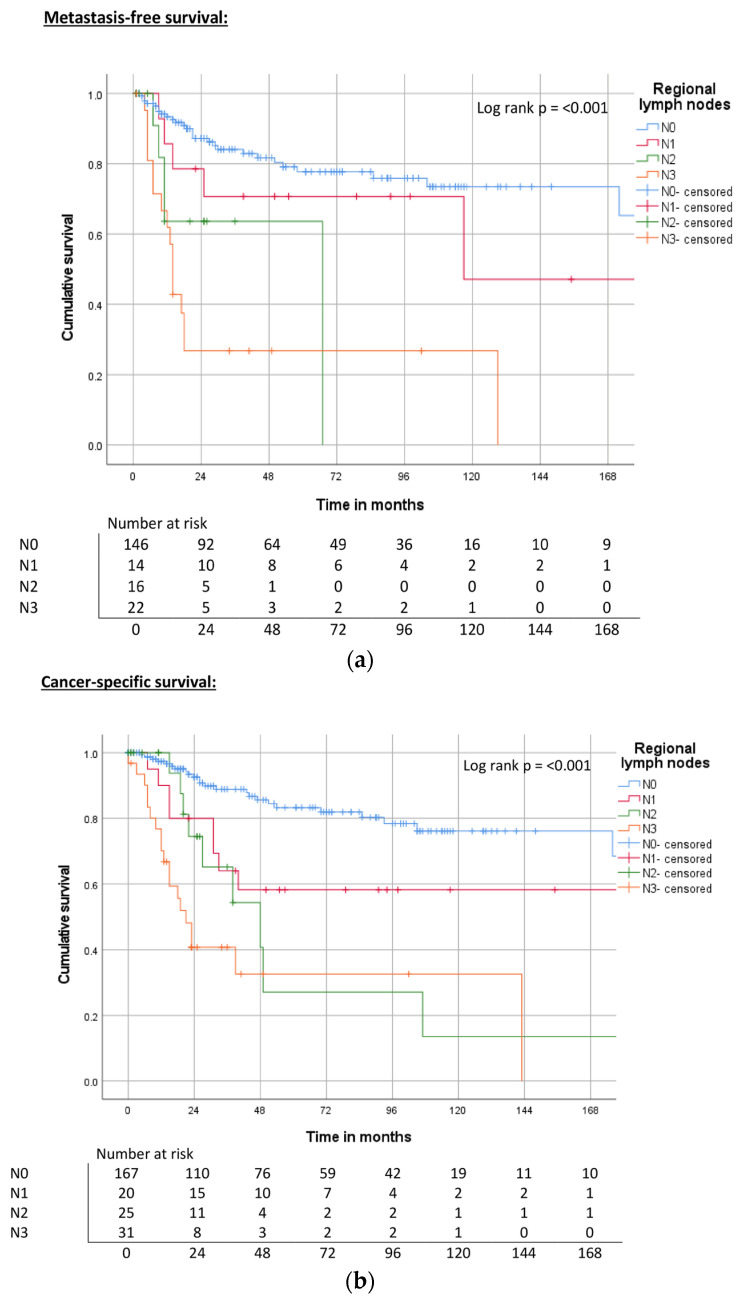
(**a**) Kaplan–Meier curve of MFS related to nodal status. (**b**) Kaplan–Meier curve of CSS related to nodal status. (**c**) Kaplan–Meier curve of OS related to nodal status.

**Figure 6 cancers-15-04748-f006:**
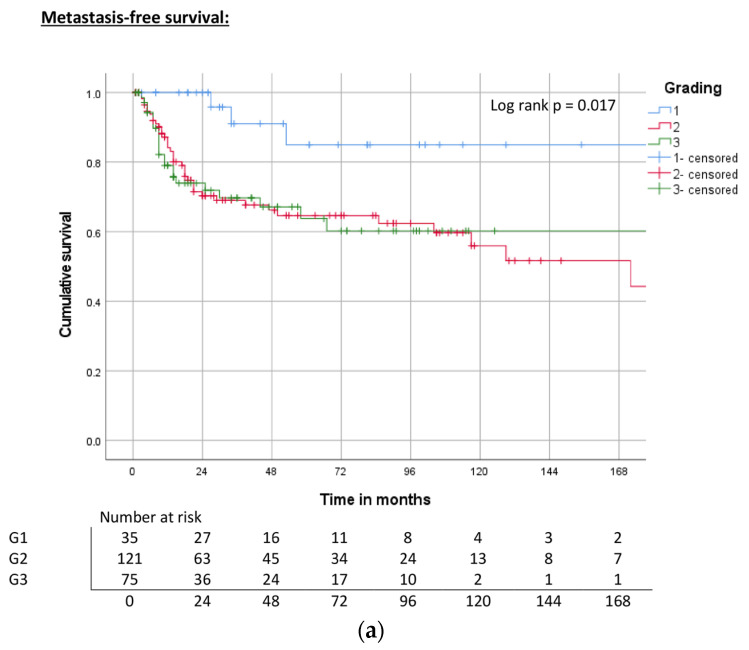
(**a**) Kaplan–Meier curve of MFS related to grading. (**b**) Kaplan–Meier curve of CSS related to grading. (**c**) Kaplan–Meier curve of OS related to grading.

**Figure 7 cancers-15-04748-f007:**
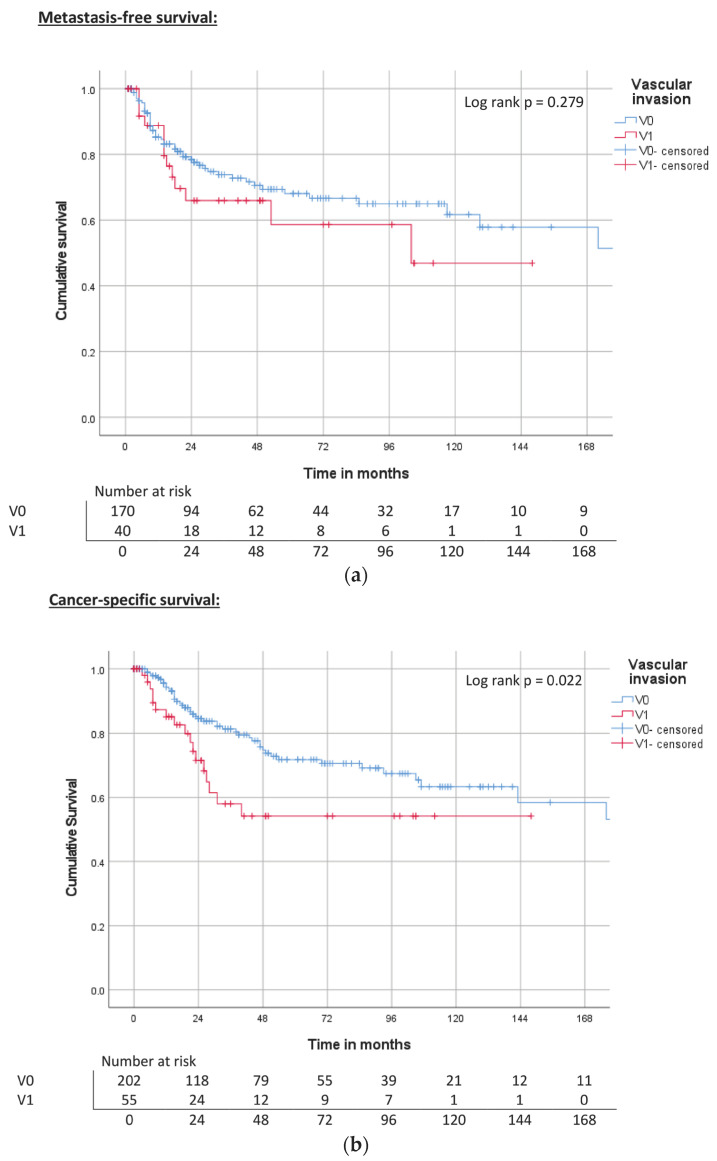
(**a**) Kaplan–Meier curve of MFS related to vascular invasion. (**b**) Kaplan–Meier curve of CSS related to vascular invasion. (**c**) Kaplan–Meier curve of OS related to vascular invasion.

**Figure 8 cancers-15-04748-f008:**
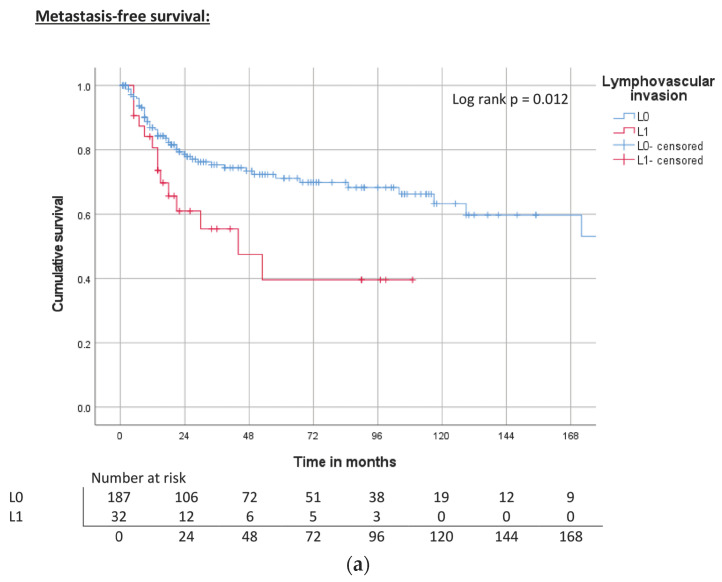
(**a**) Kaplan–Meier curve of MFS related to lymphovascular invasion. (**b**) Kaplan–Meier curve of CSS related to lymphovascular invasion. (**c**) Kaplan–Meier curve of OS related to lymphovascular invasion.

**Figure 9 cancers-15-04748-f009:**
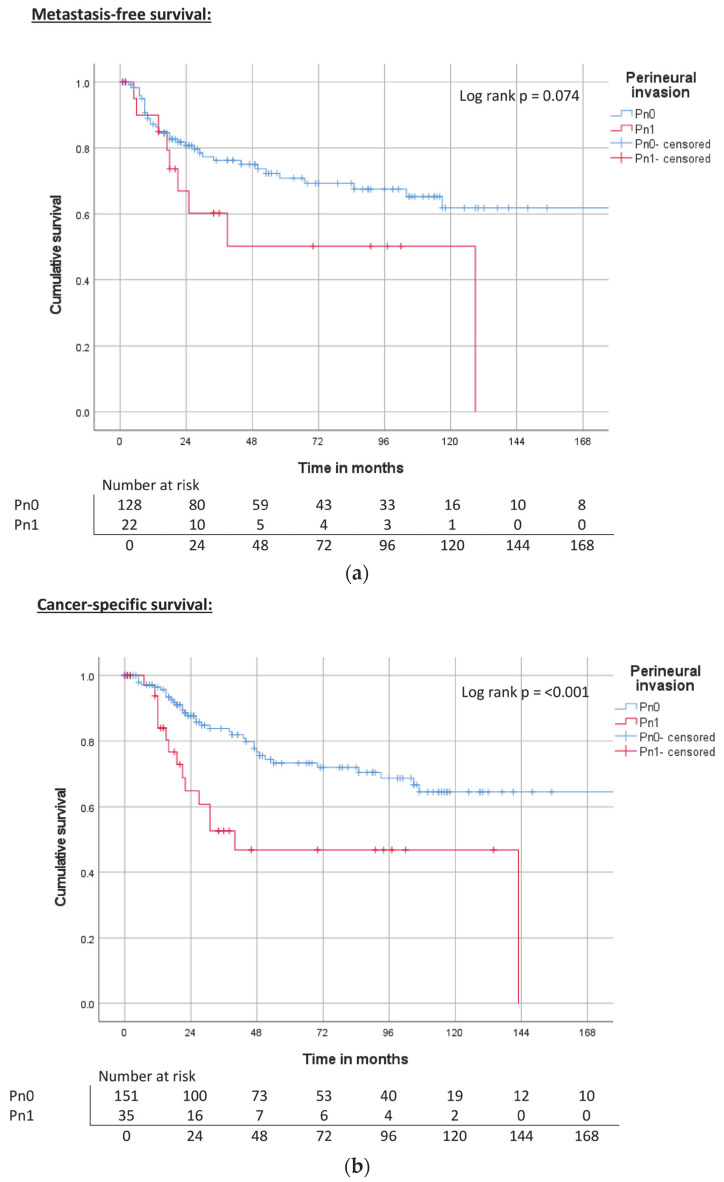
(**a**) Kaplan–Meier curve of MFS related to perineural invasion. (**b**) Kaplan–Meier curve of CSS related to perineural invasion. (**c**) Kaplan–Meier curve of OS related to perineural invasion.

**Figure 10 cancers-15-04748-f010:**
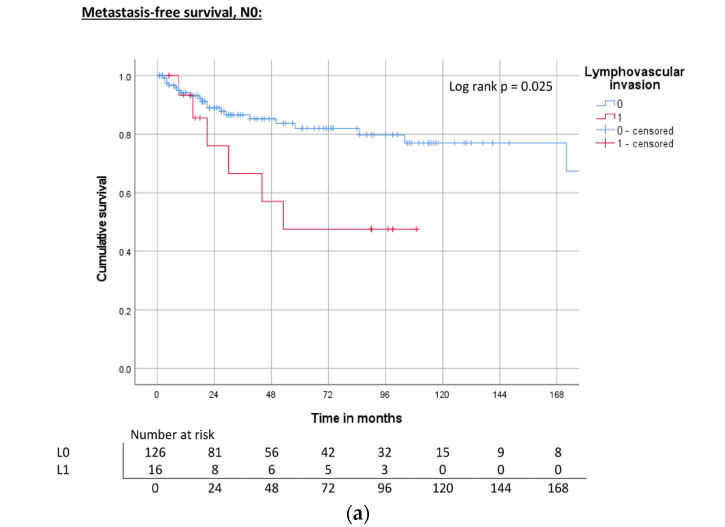
(**a**) Kaplan–Meier curve of MFS related to lymphovascular invasion in node-negative patients. (**b**) Kaplan–Meier curve of CSS related to lymphovascular invasion in node-negative patients. (**c**) Kaplan–Meier curve of OS related to lymphovascular invasion in node-negative patients.

**Table 1 cancers-15-04748-t001:** Histological subtypes.

Histological Subtype				HPV Negative		High-Risk HPV		Not Evaluable
		*n*	%	*n*	%	*n*	%	
Intraepithelial neoplasia		7	2.7	2	66.7	1	33.3	4
Non-HPV- related squamous cell carcinoma	Usual type	140	53.3	101	78.9	27	21.1	12
	Pseudohyperplastic	11	4.2	11	100	0	0.0	-
	Pseudoglandular	1	0.4	1	100	0	0.0	-
	Pure verrucous	12	4.6	9	81.8	2	18.2	1
	Carcinoma cunilatum	3	1.1	3	100	0	0.0	-
	Papillary	2	0.8	1	100	0	0.0	1
	Sarcomatoid	3	1.1	3	100	0	0.0	-
	Mixed tumors	1	0.4	1	100	0	0.0	-
HPV-related squamous cell carcinoma	Basaloid	30	11.4	6	22.2	21	77.8	3
	Papillary-basaloid	2	0.8	1	50.0	1	50.0	-
	Warty	15	5.7	12	80.0	3	20.0	-
	Warty-basaloid	35	13.3	12	34.3	23	65.7	-
	Clear cell	1	0.4	0	0.0	1	100	-

**Table 2 cancers-15-04748-t002:** Clinical and histopathological characteristics of the patient cohort.

*n* = 297		Seventh Edition		Eighth Edition	
		*n*	%	*n*	%
Primary tumor	pTis	3	1.2	3	1.2
	pT1a	65	26.2	64	25.8
	pT1b	33	13.3	34	13.7
	pT2	89	35.9	79	31.9
	pT3	51	20.6	62	25.0
	pT4	7	2.8	6	2.4
	n/a	49		49	
Regional lymph nodes	N0	167	68.7	167	68.8
	cN0	49	20.1	49	20.2
	pN0	118	48.6	118	48.6
	pN1	20	8.2	20	8.2
	pN2	25	10.3	25	10.3
	pN3	31	12.8	31	12.8
	n/a	54		54	
Distant metastasis	pM0	242	96.0	242	96.0
	pM1	10	4.0	10	4.0
	n/a	45		45	
Tumor characteristics			* **n** *	**%**	
Grading	G1		38	13.6	
	G2		150	53.6	
	G3		92	32.8	
	n/a		17		
Lymphovascular invasion (L1)	L0		220	82.1	
	L1		48	17.9	
	n/a		29		
Vascular invasion (V1)	V0		202	78.6	
	V1		55	21.4	
	n/a		40		
Perineural invasion	Pn0		151	81.2	
	Pn1		35	18.8	
	n/a		111		
**Tumor extension**					
Corpus spongiosum	No		113	47.5	
	Yes		125	52.5	
	n/a		59		
Corpus cavernosum	No		172	72.6	
	Yes		65	27.4	
	n/a		60		
Urethra	No		184	78.0	
	Yes		52	22.0	
	n/a		61		
Adjacent structures	No		229	96.6	
	Yes		8	3.4	
	n/a		60		

**Table 3 cancers-15-04748-t003:** Median and 5-year survival of the individual parameters.

Clinico-Pathological Parameters		Metastasis-Free Survival			Cancer-Specific Survival			Overall Survival		
		Median Survival Rate		5-Year Survival Rate	Median Survival Rate		5-Year Survival Rate	Median Survival Rate		5-Year Survival Rate
		Month	*p*	Survival	Month	*p*	Survival	Month	*p*	Survival
Histologic subtype	Usual type	ND	0.64	65%	ND	0.19	69%	117	0.46	59%
	Basaloid	129		76%	85		54%	39		45%
	Warty	ND		89%	176		74%	176		74%
	Warty-basaloid	ND		71%	ND		70%	57		49%
HPV	Negative	172	0.50	67%	ND	0.80	68%	118	0.394	63%
	Hr-HPV positive	ND		73%	143		75%	93		58%
Primary tumor	pT1a	172	0.003	78%	ND	0.006	85%	176	0.010	74%
	pT1b	58		44%	107		68%	107		63%
	pT2	ND		78%	ND		72%	240		64%
	pT3	104		56%	51		49%	43		39%
	pT4	21		50%	27		33%	27		33%
Regional lymph nodes	pN0, cN0	ND	<0.001	78%	214	<0.001	83%	240	<0.001	75%
	pN1	117		70%	178		58%	40		50%
	pN2	67		69%	100		28%	38		22%
	pN3	14		27%	83		32%	21		20%
Grading	G1	ND	0.017	84%	ND	0.034	85%	ND	0.025	83%
	G2	172		65%	176		65%	85		53%
	G3	ND		60%	ND		66%	107		60%
Vascular invasion	V0	ND	0.279	68%	240	0.109	72%	143	<0.001	64%
	V1	104		60%	ND		56%	31		50%
Lymphovascular invasion	L0	ND	0.012	71%	240	0.002	74%	118	0.004	67%
	L1	44		40%	40		39%	31		38%
Perineural invasion	Pn0	ND	0.074	71%	240	<0.001	74%	143	<0.001	67%
	Pn1	129		51%	40		47%	22		41%

**Table 4 cancers-15-04748-t004:** Univariable Cox Regression for the total cohort.

Clinico-Pathological Parameters		Metastasis-Free Survival		Cancer-Specific Survival		Overall Survival	
		Hazard Ratio (95% CI)	*p*	Hazard Ratio (95% CI)	*p*	Hazard Ratio (95% CI)	*p*
Primary tumor	pT1a	Reference		Reference		Reference	
	pT1b	2.9 (1.3–6.2)	**0.008**	2.2 (0.9–5.2)	0.084	1.7 (0.8–3.5)	0.158
	pT2	0.92 (0.4–2.0)	0.824	1.7 (0.8–3.6)	0.178	1.4 (0.7–2.5)	0.316
	pT3	2.7 (1.3–5.8)	**0.009**	3.2 (1.4–7.1)	**0.004**	2.8 (1.5–5.2)	**0.001**
	pT4	1.8 (0.2–13.9)	0.574	6.1 (2.1–17.8)	**0.001**	3.5 (1.3–9.4)	**0.015**
Regional lymph nodes	pN0, cN0	Reference		Reference		Reference	
	pN1	1.7 (0.7–4.4)	0.279	2.5 (1.1–5.6)	**0.023**	1.9 (0.9–3.7)	0.079
	pN2	3.8 (1.5–10.1)	**0.007**	4.3 (1.9–9.2)	**<0.001**	3.0 (1.5–5.9)	**0.001**
	pN3	7.1 (3.8–13.4)	**<0.001**	8.4 (4.5–15.7)	**<0.001**	5.2 (3.0–8.9)	**<0.001**
Grading	G1	Reference		Reference		Reference	
	G2	4.5 (1.4–14.7)	**0.012**	4.0 (1.2–13.0)	**0.02**	3.0 (1.3–7.0)	**0.010**
	G3	4,7 (1,4–15,7)	**0.012**	4.2 (1.3–14.0)	**0.019**	2.7 (1.1–6.5)	**0.026**
Vascular invasion	V0	Reference		Reference		Reference	
	V1	1.4 (0.8–2.6)	0.284	1.9 (1.1–3.3)	**0.025**	1.6 (1.0–2.7)	0.054
Lymphovascular invasion	L0	Reference		Reference		Reference	
	L1	2.1 (1.2–3.9)	**0.015**	2.6 (1.5–4.5)	**0.001**	2.2 (1.4–3.5)	**0.001**
Perineural invasion	Pn0	Reference		Reference		Reference	
	Pn1	1.9 (0.9–4.1)	0.081	2.8 (1.5–5.1)	**0.001**	2.3 (1.3–4.0)	**0.004**

**Table 5 cancers-15-04748-t005:** Multivariable Cox Regression for the total cohort.

Clinico-Pathological Parameters		Metastasis-Free Survival		Cancer-Specific Survival		Overall Survival	
		Hazard Ratio (95% CI)	*p*	Hazard Ratio (95% CI)	*p*	Hazard Ratio (95% CI)	*p*
Primary tumor	pT1a	Reference		Reference		Reference	
	pT1b	7.8 (1.6–13.9)	**0.017**	n.s	0.939	n.s	0.703
	pT2	0.8 (0.3–1.9)	0.097	n.s	0.827	n.s	0.970
	pT3	2.2 (0.9–5.5)	0.324	n.s	0.675	n.s	0.895
	pT4	1.9 (0.2–14.7)	0.475		**0.035**	n.s	0.134
Regional lymph nodes	pN0, cN0	Reference		Reference		Reference	
	pN1	1.8 (0.7–4.8)	0.216	2.2 (0.8–6.0)	0.138	2.1 (0.9–5.0)	0.082
	pN2	3.9 (1.5–10.2)	**0.007**	4.9 (2.1–11.6)	**<0.001**	4.2 (2.0–8.7)	**<0.001**
	pN3	6.8 (3.5–13.0)	**<0.001**	5.9 (2.7–12.8)	**<0.001**	4.2 (2.1–8.4)	**<0.001**
Grading	G1	Reference		Reference		Reference	
	G2	n.s	0.221	n.s	0.510	n.s	0.410
	G3	n.s	0.959	n.s	0.864	n.s	0.891
Vascular invasion	V0	Reference		Reference		Reference	
	V1	n.s	n.s	n.s	0.430	n.s	n.s
Lymphovascular invasion	L0	Reference		Reference		Reference	
	L1	1.6 (0.8–3.3)	0.346	2.7 (1.4–5.4)	**0.005**	2.3 (1.3–4.4)	**0.007**
Perineural invasion	Pn0	Reference		Reference		Reference	
	Pn1	n.s	n.s	n.s	0.624	n.s.	0.628

**Table 6 cancers-15-04748-t006:** Univariable Cox Regression for nodal negative patients.

Clinico-Pathological Parameters		Metastasis-Free Survival		Cancer-Specific Survival		Overall Survival	
		Hazard Ratio (95% CI)	*p*	Hazard Ratio (95% CI)	*p*	Hazard Ratio (95% CI)	*p*
Primary tumor	pT1a	Reference		Reference		Reference	
	pT1b	1.7 (0.5–5.7)	0.367	2.2 (0.5–9.4)	0.280	2.1 (0.8–5.8)	0.141
	pT2	1.0 (0.4–2.4)	0.927	1.7 (0.6–5.1)	0.337	1.0 (0.5–2.4)	0.922
	pT3	1.4 (0.4–4.6)	0.577	2.7 (0.7–10.1)	0.147	2.7 (1.1–6.5)	**0.031**
	pT4	2.5 (0.3–20.0)	0.382	6.3 (1.2–32.5)	**0.030**	2.7 (0.6–12.2)	0.200
Grading	G1	Reference		Reference		Reference	
	G2	3.5 (0.8–15.0)	0.092	2.4 (0.6–10.6)	0.244	2.8 (0.8–9.2)	0.095
	G3	2.5 (0.5–12.4)	0.264	2.3 (0.5–11.4)	0.292	2.8 (0.8–9.9)	0.115
Vascular invasion	V0	Reference		Reference		Reference	
	V1	1.0 (0.3–3.5)	0.966	0.7 (0.2–3.0)	0.645	0.9 (0.3–2.6)	0.855
Lymphovascular invasion	L0	Reference		Reference		Reference	
	L1	2.8 (1.1–6.9)	**0.032**	2.0 (0.8–5.5)	0.167	1.6 (0.7–3.7)	0.247
Perineural invasion	Pn0	Reference		Reference		Reference	
	Pn1	1.7 (0.4–7.3)	0.491	0.8 (0.1–5.9)	0.816	0.5 (0.1–3.8)	0.523

## Data Availability

The data can be shared upon request.

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
