# Peer review of "Evaluation of Prognostic Parameters to Identify Aggressive Penile Carcinomas"

_cancers, 2023, doi:10.3390/cancers15194748_

Round 1
Reviewer 1 Report
The current article holds significant relevance for urologists. Penile cancer (PC) is characterized by a relatively low incidence, resulting in a shortage of sizable patient cohorts and restricted data availability for comprehensive analysis. Even meta-analyses have encountered substantial challenges due to the limited number of PC cases.
This research sheds light on the identification of risk and prognostic factors associated with penile cancer. One intriguing question arises from the findings: why does pT1b exhibit a worse prognosis when compared to stages T2 to T4? It is imperative to explore whether this discrepancy could be attributed to differences in treatment protocols or variations in lymph node dissection templates. Therefore, it would be beneficial to provide a detailed description of the treatment protocols employed in the Materials and Methods section and engage in a comprehensive discussion regarding this specific issue.
Additionally, the presence of positive surgical margins may represent another potential risk factor for adverse outcomes in PC patients. It is crucial to ascertain whether data pertaining to surgical margin status is available and, if so, to include it in the analysis.
Furthermore, understanding the treatment landscape is essential. Were the patients in the study cohort subjected to adjuvant or salvage chemotherapy? If so, it is essential to elucidate the specific treatment regimens and protocols employed. A comprehensive description of the treatment strategies will enhance the readers' understanding of the therapeutic interventions utilized in the study.
In summary, addressing these questions and providing detailed information on treatment protocols, surgical margins, and chemotherapy regimens will contribute to a more comprehensive and informative research article, making it a valuable resource for urologists and researchers in the field.
Author Response
Thank you for your positive review and suggestions for improvement. I completely agree with you that both treatment protocols, chemotherapy regimens and surgical margins during surgery can have a significant impact on prognosis in penile cancer. Especially in the pT1b stage, adequate lymph node management is of extraordinary importance. A weakness of this study, as already mentioned in the limitations (“Moreover, patients were evaluated over a period of almost 30 years; therefore, a wide variety of therapy concepts were applied that could not be considered in this study.”), is certainly the large study period of over 30 years as well as the large number of participating nationalities and institutes. This leads to a large number of different therapy protocols, some of which can no longer be traced back with certainty, so that it does not appear possible to include the therapies performed in this study retrospectively. However, as correctly pointed out, the therapy regimen is important for the prognosis estimation, so that it will be our goal to prospectively record it in the future to evaluate the influence of the therapies performed as well as the surgical margins on the prognosis. Nevertheless, we think that this study provides important results for prognostic estimation and points out possible weaknesses of the current TNM classification.
Reviewer 2 Report
Authors describe a large series of penile carcinoma, trying to identify prognostic factors, which endpoint is cancer specific survival, metastasis-free survival, and overall survival.
The casuistic is interesting and contemplates different countries in Europe and Asia.
Some aspects must be considered
- Why authors don’t use only the classification and recommendations of WHO/IARC 5th edition?
- Authors mention lymphovascular and vascular tumor embolization. Don’t they meant to describe lymphatic and vascular?
- How many tissue cores of each sample were put in the TMA? What is the size of tumor spots?
- “In situ carcinoma” is not recommended by WHO/IARC publication (5th edition). The better nomenclature would be “intraepithelial neoplasia”.
- How authors have classified the penile tumors regarding HPV? Considering molecular, immunohistochemistry or either one?
- It would be better to reduce the number of tables and graphics.
Author Response
Thank you for your review and suggestions for improvement. In the following, I would like to briefly address the individual points of your review.
- Why authors don’t use only the classification and recommendations of WHO/IARC 5th edition?
- The WHO classification of penile carcinoma was updated to the current 5th edition in 2022. The evaluation and updating of our database has already taken place since 2018, so that all tumors and histological subtypes were categorized according to the then current TNM classification. However, a further update to the current WHO classification is planned further down the line with prospective data collection
- Authors mention lymphovascular and vascular tumor embolization. Don’t they meant to describe lymphatic and vascular?
- In our work, we use the term lymphovascular instead of lymphatic, which is also recognized in the literature, to clarify the distinction between lymph node and lymphatic vessel infiltration. We added also the TNM classification (L1, V1) in Table 2.
- How many tissue cores of each sample were put in the TMA? What is the size of tumor spots?
- Depending on availability, 2-3 tumor samples from the center, tumor front, and lymph node metastases were used for TMAs. The size of the tumor spots was 1.5 mm. This is now stated in the MS on page 3. Here the sentence „Tumor samples from the center, tumor front, and lymph node metastases with a spot size of 1.5 mm each were used for tissue microarry construction.“ was added.
- “In situ carcinoma” is not recommended by WHO/IARC publication (5th edition). The better nomenclature would be “intraepithelial neoplasia”.
- Thank you for this helpful hint, I have deleted the term carcinoma in situ in the corresponding table.
- How authors have classified the penile tumors regarding HPV? Considering molecular, immunohistochemistry or either one?
- As outlined in the Materials and Methods section, DNA isolation from FFPE tissues was performed for HPV determination. Furthermore, to evaluate HPV-induced tumorigenesis, p16INK4a was determined by immunohistochemistry using a monoclonal antibody against p16INK4a (Abcam, clone 1D7D2A1). The cases were considered HPV positive if both detection of HPV DNA occurred and overexpression of p16 was detected by immunohistochemistry.
- It would be better to reduce the number of tables and graphics.
- It is correct that many graphs and tables were used for this work. However, these are necessary for the completeness of this work. Possibly, in coordination with the publisher MDPI, some graphs can be moved to the supplement.